# Thermal Analysis of a MEMS-Based Self-Adaptive Microfluidic Cooling Device

**DOI:** 10.3390/mi12050505

**Published:** 2021-04-30

**Authors:** Gonzalo Sisó, Joana Rosell-Mirmi, Álvaro Fernández, Gerard Laguna, Montse Vilarrubi, Jérôme Barrau, Manuel Ibañez, Joan Rosell-Urrutia

**Affiliations:** 1Dynamic Systems Applied to Solar Energy Research Group, University of Lleida, Avda Jaume II 69, 25001 Lleida, Spain; gonsi391@gmail.com (G.S.); alvarofv4@gmail.com (Á.F.); montse.vilarrubi@udl.cat (M.V.); jerome@macs.udl.cat (J.B.); m.ibanez@macs.udl.cat (M.I.); rosell@macs.udl.cat (J.R.-U.); 2Centre Internacional de Mètodes Numèrics en Enginyeria, Building Energy and Environment Group, CIMNE–Lleida, Pere de Cabrera 16, Office 2G, 25001 Lleida, Spain; glaguna@macs.udl.cat

**Keywords:** microfluidic cell, self-adaptive valve, cooling device

## Abstract

This study presents a thermal analysis of a temperature-driven microfluidic cell through a nonlinear self-adaptive micro valve that provides the mechanisms for the system to maintain a given critical temperature in an efficient way. For the description of the dynamics of the microfluidic cell, a system of two ordinary differential equations subjected to a nonlinear boundary condition, which describes the behavior of the valve, is proposed. The solution of the model, for determined conditions, shows the strong nonlinearity between the overall thermal resistance of the device and the heat flux dissipated due to the action of the thermostatic valve, obtaining a variable thermal resistance from 1.6 × 10^−5^ to 2.0 × 10^−4^ Km^2^/W. In addition, a stability analysis of the temperature-driven microfluidic cell is presented. The stability of the device is essential for its proper functioning and thus, to prevent its oscillating behavior. Therefore, this work focuses on assessing the range of design parameters of the self-adaptive micro valve to produce a stable behavior for the entire system. The stability analysis was performed by studying the linear perturbation around the stationary solution, with the model solved for various heat flows, flow rates, and critical temperatures. Finally, a map of the design parameters space, which specifies the region with asymptotic stability, was found. In this map, the critical temperature (temperature at which the valve initiates the buckling) plays and important role.

## 1. Introduction

Physicists agree with Moore’s law and its 18-month doubling transistor density trend, where a transistor has about a five-year life before encountering size shrinking limitations imposed by the laws of thermodynamics and quantum physics. Today’s advanced ICT trends involve further integration at the die and package level. Since over-heated dies are known to consume more power and therefore increase heating, failing to find an appropriate cooling solution for advanced micro- and nanoelectronics systems is one of the major challenges of the ICT community.

In recent years, many works in the field of heat dissipation have focused on the development of microfluidic cooling devices [1,2] as a means to achieve a reduction in energy consumption. Barrau et al. [3,4,5] and Rubio Jimenez et al. [6] proposed different ways of reaching more uniform temperatures of heated zones by using a hybrid jet impingement/micro-channel cooling device and a micro-pin fin heat sink with variable fin density. Along this line, a cooling scheme consisting of an array of microfluidic cells with adaptive valves capable of maintaining a surface temperature below a critical point by locally modulating flow rate in response to changes in heat load, was investigated by [7,8,9]. The concept of self-adaptive valve was previously assessed by McCarthy et al. [10], and the authors demonstrated that the mechanism of thermal buckling in a clamped beam fabricated over a slot was able to modulate the flow rate as a function of the valve temperature. Additionally, Li et al. [11] reported a self-adaptive microchannel with a thermal-sensitive nanocomposite hydrogel. The hydrogel acts as a smart valve and tailors the flow rate from 0.14 to 0.46 mL/s based on heat load variation. Compared with conventional channels, heat flux of 100 W/cm^2^ can be extracted with similar temperature rise while the coefficient of performance (COP) is improved by an order of magnitude.

Other types of temperature-driven self-adaptive valves have been reported in the literature. Wang et al. [12] used a wax thermostatic valve in a fluid loop for spacecraft thermal management, demonstrating that it is possible to obtain quick control of the instabilities originated in a thermostat valve. These instabilities, which appear in the thermostatic control, are an important issue that causes overheating, oscillatory behavior, and even chaos, and are known as the “thermostat problem” [13,14,15,16]. This problem derives mainly from the delay between the changes in both the coolant flow rate and the temperature of the solid part that governs the thermostatic valve, and is aggravated in a system based on a matrix of microfluidic cells with self-adaptive valves [8] due to its increased complexity. In this case, the stability of a simple cooling cell is essential to ensure its correct operation. For this reason, this work focuses on assessing the range of design parameters of a self-adaptive micro valve that ensures a stable behavior of the microfluidic cell and the self-adaptive valve.

A first step in the evaluation of the stability of a dynamic system is to analyze the linear stability of stationary solutions. For example, Farjas et al. [17] presented a reduced ordinary differential equation system (ODEs) for a set of partial differential equations (PDE) for heat conduction with the second law of Fourier as the main equation in order to assess the linear stability of the stationary solutions. The linearization of the ODEs was effectuated by analyzing the first-order perturbations over the stationary solutions, and the stability of the stationary solution was related to the eigenvalues and eigenvectors of the linear matrix of the transformation. The stationary solution is stable if all the eigenvalues have a negative real part. Moreover, it is possible to identify two paths to instability by analyzing the changes in the eigenvalues of the system [18]. In the first case (Figure 1, route a), one eigenvalue changes from a real negative value to a real positive value (saddle-node bifurcation). In the second case, a couple of real negative eigenvalues present imaginary parts different than zero while the stationary point is still stable (oscillatory transition, Figure 1, route b), and then the couple of imaginary eigenvalues change the real part from a negative to a positive value (Hopf bifurcation Figure 1, route c).

In this work, a reduced system of two differential equations for the microfluidic cell dynamics description, based on the balance of energy rates in each of the components, is proposed. The equations describe the temporal evolution of two lamped variables: the mean temperature of the solid part of the cell, and the mean temperature of the liquid within the cell. The stationary solutions have been analyzed and the convection coefficients defined, so the ODE system satisfies the original PDE stationary solutions. The stationary and dynamic behavior of the new system, noun as extended ODEs and based on the microfluidic cell (Equations (1) and (2)) and the self-adaptive valve equations (Equations (7)–(9)), is also evaluated. The linear stability of the extended ODEs is assessed through a new linear equation system from the analysis of the first-order perturbation over the stationary solutions, thus providing the eigenvalues and eigenvectors that characterize the dynamics of the system around stationary points. These results are used to define the stable zone on the parameter space and identify the parameter ranges where self-induced damping oscillations are found.

## 2. Modeling

The microfluidic cell used in this work [8] is made of silicon with external dimensions of 1.0 × 1.2 × 0.3 mm. The internal design consists of six micro channels with different lengths (Figure 2) that allow the heat exchange area along the flow path to be increased. Both the inlet and the outlet are on the opposite side of the heated surface. The micro valve, located on the outlet slot (Figure 2), consists of a doubly-clamped beam (1.0 mm long, 2 µm thick, and 100 µm wide).

First, the mathematical model that describes the thermal behavior of the microfluidic cell without the thermostatic valve is analyzed. Then, the thermal behavior of the valve is assessed and finally, the evaluation of the extended model is done.

### 2.1. Microfluidic Cell Model

The proposed mathematical system for the evaluation of the microfluidic cell is based on the two-node one-segment model extracted using the improved transfer function method described by W. Kong [19] and obtained through an energy balance. A constant heat flux *q* (W/m^2^) has been applied at the bottom part of the microfluidic cell:(1)$$C_{b}\frac{dT_{b}}{dt}=q+U_{L}\left(T_{a}-T_{b}\right)+U_{b}\left(T_{f}-T_{b}\right)$$
(2)$$C_{f}\frac{dT_{f}}{dt}=-U_{f}\left(T_{f}-T_{in}\right)+U_{b}\left(T_{b}-T_{f}\right)$$
where *T_b_* is the mean temperature of the solid matter of the microfluidic cell and *T_f_* is the mean coolant temperature. *T_a_* and *T_in_* are the ambient and input coolant temperatures. *U_L_* is the overall coefficient of heat loss, *U_b_* is the convective coefficient between the microcell walls and the fluid, and *U_f_* is the advection coefficient of the fluid, defined as: (3)$$U_{f}=\frac{C_{f}\Gamma Q_{f}}{L\;S}\;$$

The Γ factor, defined in Hilmer et. al. [20] by Equation (4), has a value between 1 and 2 and varies strongly with the temperature distribution inside the cell. When *Q_f_* is high, the stationary temperature distribution is almost linear and (*T_out_* − *T_in_*) is twice the value of (*T_f_* −*T_in_*), so Γ takes a value of 2. With a reduced *Q_f_*, (*T_out_* − *T_in_*) is nearly the same value of (*T_f_* − *T_in_*), then Γ has a value close to 1. Owing to the definition of the Γ factor in Equation (4), (*T_f_* − *T_in_*) is used instead of (*T_out_* −*T_in_*) in Equation (2):(4)$$T_{out}-T_{in}=\Gamma \left(T_{f}-T_{in}\right)\;$$

The equation system, Equations (1) and (2), has five parameters to calibrate: *C_b_*, *C_f_*, *U_L_*, *U_b_*, and *U_f_*. To determine the values of thermal capacities, (*C_b_* and *C_f_*) the values of density, specific heat, and volumes of the constituent materials of microfluidic cells (silicon and water, respectively) were used (Table 1). The parameters *U_L_*, *U_b_*, and *U_f_*, were extracted by comparing the stationary solution of the PDE model for different coolant flow rates and heat fluxes. The expressions for *U_b_* and *U_f_* used were defined as:(5)$$U_{b}\left(Q_{f},q\right)=\left(\frac{c_{1}\cdot q^{2}+c_{2}\cdot q+c_{3}}{S^{2}}\right)\cdot Q_{f}^{2}+\left(\frac{m_{1}\cdot q+m_{2}}{S}\right)\cdot Q_{f}+r_{1}\;$$
and
(6)$$U_{f}\left(Q_{f}\right)=\frac{n_{1}}{S}\cdot Q_{f}+n_{2}$$

### 2.2. Self-Adaptive Valve Condition

The micro valve position and buckling beam action are shown in Figure 3. When the temperature in the microfluidic cell is higher than the critical temperature, the valve buckles, increasing the flow rate, and therefore the temperature of the cell decreases. As a result, in situations of low thermal load, the adaptive scheme reduces the coolant flow rate, leading to higher surface temperatures. This behavior avoids overcooling under time-dependent heat load scenarios and saves pumping energy.

McCarthy [10] described the behavior of a nonlinear thermal valve by means of a set of transcendental equations using the dimensionless temperature *θ* and a flow rate per unit pressure drop  ∅. It was solved numerically for a given eccentricity ratio (0.2) and was written in terms of the exponential function:(7)$$\emptyset =K_{1}\left(e^{\tau \theta^{ex}}-1\right)$$
where the dimensionless temperature is:(8)$$\theta =\frac{T_{b}-T_{ref}}{T_{crit}-T_{ref}}$$
where *T_crit_* and *T_ref_* are part of the valve design. *T_crit_* is the critical temperature at which the valve initiates the buckling, and *T_ref_* is the temperature at which the beam micro valve has no thermal stress. Working under a constant Δ*P,* the flow rate is assessed as:(9)$$Q_{f}=\emptyset \cdot Q_{f}{}_{crit}$$
where $Q_{f}{}_{crit}$ represents the water flow rate when the buckling begins (*θ* = 1).

### 2.3. Microfluidic Cell Model with Self-Adaptive Valve Condition: The Extended Model

The previously presented set of Equations ((1)–(3), (5)–(7), (9)) define the microfluidic cell model with self-adaptive valve condition, noun from here as the extended model.

By setting the time derivative equal to zero in Equations (1) and (2), the stationary solution for the microfluidic cell model is obtained as:(10)$$T_{b}=\frac{U_{b}\cdot \left(U_{f}\cdot T_{in}+U_{L}\cdot T_{a}+q\right)+T_{a}\cdot U_{L}\cdot U_{f}+q\cdot U_{f}}{U_{b}\cdot \left(U_{f}+U_{L}\right)+U_{L}\cdot U_{f}}$$
and
(11)$$T_{f}=\frac{U_{b}\cdot \left(U_{f}\cdot T_{in}+U_{L}\cdot T_{a}+q\right)+T_{in}\cdot U_{L}\cdot U_{f}}{U_{b}\cdot \left(U_{f}+U_{L}\right)+U_{L}\cdot U_{f}}$$

### 2.4. Linearized System

To carry out the stability study, the linearized equation system around a stationary solution was established.

A small disturbance Δ*T* was added around the stationary solution *T_bs_* and *T_fs_*: *T_b_* = *T_bs_* + Δ*T_b_**T_f_* = *T_fs_* + Δ*T_f_*,(12)
and then:(13)$$\begin{array}{c} U_{f}=U_{fs}+{\frac{dU_{f}}{dT_{b}}\rfloor }_{s}\cdot \Delta T_{b}=U_{fs}+{U^{\prime }}_{f}\cdot \Delta T_{b}\\ U_{b}=U_{bs}+{\frac{dU_{b}}{dT_{b}}\rfloor }_{s}\cdot \Delta T_{b}=U_{bs}+U\prime_{b}\cdot \Delta T_{b} \end{array}$$

The new linear model obtained by substituting the Equations (12) and (13) into Equations (1) and (2) is:(14)$$\Delta {\dot{T}}_{b}=a_{1}\cdot \Delta T_{b}+a_{2}\cdot \Delta T_{f}$$
(15)$$\Delta {\dot{T}}_{f}=b_{1}\cdot \Delta T_{b}+b_{2}\cdot \Delta T_{f}.$$
with
(16)$$a_{1}=-\frac{1}{C_{b}}\left(\left(U_{L}+U_{bs}\right)+\left(T_{bs}-T_{fs}\right)\cdot U_{b}^{\prime }\right)$$
(17)$$a_{2}=\frac{U_{bs}}{C_{b}}\;$$
and
(18)$$b_{1}=\frac{1}{C_{f}}\cdot \left[\left(U_{bs}-\left(T_{fs}-T_{in}\right)\cdot U_{f}^{\prime }\right)+\left(T_{bs}-T_{fs}\right)\cdot U_{b}^{\prime }\right]$$
(19)$$b_{2}=-\frac{1}{C_{f}}\left(U_{fs}+U_{bs}\right)$$

The system can be placed in matrix form:(20)$$\left(\begin{array}{c} \dot{\Delta Tb}\\ \dot{\Delta Tf} \end{array}\right)=A\left(\begin{array}{c} \Delta Tb\\ \Delta Tf \end{array}\right)$$
with
(21)$$A=\left[\begin{array}{cc} a_{1} & a_{2}\\ b_{1} & b_{2} \end{array}\right]$$
with the characteristic polynomial:(22)$$\lambda^{2}-\left(Tr\left(A\right)\right)\lambda +Det\left(A\right)=0$$
where Tr (*A*) and Det (*A*) are the trace and the determinant of the *A* matrix, respectively. The eigenvalues are the roots of Equation (22), which is solved as:(23)$$\lambda =\frac{-\left(a_{1}+b_{1}\right)\pm \sqrt{{\left(a_{1}-b_{2}\right)}^{2}+4a_{2}b_{1}}}{2}$$

Disc (*A*) is the discriminant of Equation (23).

For the stability study, the change of sign in the determinant and discriminant was analyzed. When the determinant is zero, at least one of the eigenvalues is zero (Equation (22)) and this indicates a change in the sign of eigenvalues. When the discriminant is negative, the square root in Equation (23) will provide a couple of conjugated solutions both with real and imaginary parts.

## 3. Results and Discussion

### 3.1. Microfluidic Cell Model

The functional dependences of Equations (5) and (6) were validated by comparing the stationary temperatures (Figure 4a) obtained with the proposed microfluidic cell model (ODEs, Equations (1)–(6)) and the reference partial differential equations model (PDEs). The PDE model used in the comparison was a model that coupled the transfer of heat into weakly compressible fluids (density depends on temperature but not on pressure), in contact with solids. The interface of non-isothermal laminar fluid was used in conjunction with conjugated heat transfer. Using the geometry shown in Figure 2 as a domain, the PDE model was solved with Comsol Multiphysics. A grid independency test has been carried out to ensure the independency of results from mesh size, obtaining a final mesh of 7.5 × 10^5^ elements (free tetrahedral elements), that have an error lower than 1% with respect to smaller grids. The boundary conditions used were symmetry for all side faces, known heat flux *q* for the bottom face and ambient convection for the top face. Different simulations from a finite element analysis solver were used. The coolant flow and dissipated power were varied between 1.5 × 10^−9^ and 1.5 × 10^−8^ m^3^/s, and 10.0 W/cm^2^ and 300.0 W/cm^2^, respectively. The results of the calibration of the Equations (5) and (6) are shown in Table 2 and the comparison between the stationary temperatures *T_b_* and *T_f_* for all the combinations performed are shown in Figure 4a. The average quadratic error committed was 4.8 × 10^−3^ K for *T_b_* values and 3.9 × 10^−3^ K for *T_f_* values.

The comparison between the dynamic behavior of both models for 4 different transitions is shown in Figure 4b. In this figure, appears the time evolution of Tb for an initial thermal flow of q_0_ = 300 W/cm^2^ and flow rate of Q_0_ = 5·10^−8^ m^3^/s. The 4 transitions were: *t* = 0 s initial heat up to q_0_, *t* = 0.1 s change to q_1_ = 150 W/cm^2^, *t* = 0.2 s again q_0_ = 300 W/cm^2^, and t = 0.3 reduction of the coolant flow rate Q = 3/4 Q_0_. It can be seen that the difference between the temperatures predicted by the two models was, in any case, lower than 0.6 K and the average quadratic error was 0.9 K. Up to 20 dynamic simulations similar to those presented in Figure 4 were performed with different values of Q and *q*. In all of them, the RMSE value was less than 1.5 K.

### 3.2. Self-Adaptive Valve Condition

The parameters of Equation (7) were calibrated by reducing the root mean square error for a given micro valve, and the resultant values are listed in Table 3.

The value of the dimensionless flow rate as a function of dimensionless temperature for the McCarthy equation and exponential curve (Equation (7)) are graphically shown in Figure 5. The root mean square error committed in the approximation was 0.016(-).

### 3.3. Stationary Solution of Microfluidic Cell Model with Self-Adaptive Valve Condition (Extended Model)

Considering the equations of heat exchange coefficients (Equations (5) and (6)) and the valve (Equations (7) and (9)), a stationary solution for the microfluidic cell with the adaptive valve was found. The system is not analytically solvable and must be solved by an optimization method. In this case, the reduced generalized gradient method was used. Stationary solutions were found for several values of heat flux (10–300 W/cm^2^). The parameters *T_crit_*, *T_ref_*, and *Q_f,crit_* (Table 1) were established with the criteria of limiting temperature T_b_ to 80 °C even at higher heat fluxes.

Figure 6 and Figure 7 show the results of the stationary solution of the extended model in terms of temperature T_b_ and thermal flux *q*.

The curve in Figure 6a is not a straight line because of the influence of the thermal valve nonlinear behavior. In addition, Figure 6b shows that the flow rate increases with the cell temperature. The previously mentioned nonlinear behavior also appears in the relationship between the cell temperature, the coolant flow rate, and thermal resistance with the heat flux (Figure 7). The thermal resistance, calculated as R_t_ = (T_b_-T_in_)/*q*, shows a strong non-linearity. This fact is the key to its use in this type of application, ultimately the temperature of the valve regulates the effective resistance of the device. As the temperature increases, the flow rate increases and the thermal resistance values is significantly reduced to 1.6 × 10^−5^ K·m^2^/W.

Additionally, at low temperatures, the flow rate is small, and the energy consumption related to the refrigerant system is therefore small. Significant energy costs appears only when the system needs to dissipate high thermal fluxes.

### 3.4. Stability Analysis of Microfluidic Cell Model with Self-Adaptive Valve Condition

With the aim of identifying the instabilities that may appear in the working range, the calculation of the determinant and the discriminant values of linearized system were performed for several heat fluxes from 10 to 300 W/cm^2^ (Figure 8).

The obtained results show that the determinant always presents positive values and increases with temperature rise. Therefore, it does not cause instabilities despite the discriminant start from positive values and changes of sign at a certain heat flow value (*q**_c_* = 248 W/cm^2^). When the discriminant has a positive sign, both eigenvalues are real and have negative signs, so they are then stable points. However, when the discriminant becomes negative, the eigenvalues change to complex numbers with imaginary parts (Figure 9 and Figure 10). In these situations, self-oscillatory behavior is observed, i.e., a spiral trajectory in the variables space; however, both eigenvalues have negative real parts; therefore, the orbit converges towards a stationary solution.

The study of instabilities was extended by modifying the critical flow and temperature valve-design parameters. The Q*_f,crit_* and *T_crit_* values were varied between 1.0 and 6.0 × 10^−8^ m^3^/s and 305 and 315 K, respectively. In addition, the heat flux q range of 10–300 W/cm^2^ was used to simulate the new working conditions of the valve.

It was observed that, in the configuration of the valve presented in this work, the slope of the characteristic curve was always positive and sign changes were only seen in the discriminant. Therefore, we focused on discriminant sign changes.

For each assessed value of T_crit_ (Figure 11), the points at which the discriminant of the linear transformation became zero were determined.

For each *T_crit_*, the line shown in the graph separates the asymptotically stable region (left) from the self-oscillatory region (right). In this last region, the device has a damped oscillatory response that should be avoided to obtain a self-adaptive behavior without fluctuations. The *Q_f,crit_* and *T_crit_* values are design variables because of their strong dependence on geometrical dimensions and the characteristics of the material that form the valve. The *Q_f,crit_* value also depends on the pressure difference of the coolant fluid to which the valve is subjected.

An outstanding result was that the stability zone increased with *T_crit_* value, meaning that the average working temperature of the device also increased.

To obtain a broader knowledge of the behavior of the device in other conditions, we performed a second parametric study and analyzed its response by varying another significant parameter in the dynamic evolution, such as the slope of the characteristic curve of the thermostatic valve through an analysis of the variation of the coolant velocity with temperature (Figure 12). These changes would represent not only a design parameter change but a change in the physical structure of the valve. The new parameter is related to the stability of the system through the expression of *U’_f_* and *U’_b_* of the linearized model (Equations (16) and (18)).

It can be seen, in Figure 12, the evolution of both the determinant and the discriminant values by varying the derivative of the fluid velocity with temperature. Figure 13 show the values of the real part of the eigenvalues as a function of this derivative. We can clearly observe (Figure 12) three distinct intervals according to the type of stability they represent:(1)The first interval, with derivative values of −0.1 to −0.01 m/s·K, where the discriminant has a positive value and the determinant a negative value. This means that at least one of the real parts of the eigenvalues is positive and, therefore, the stationary point is asymptotically unstable (saddle type).(2)The second interval, with derivatives values from −0.01 to 0.04 m/s·K, where the determinant changes its sign and becomes positive. Here, the stationary points are characterized by asymptotically stable points.(3)The third interval, with values of the derivative from 0.04 to 0.1 m/s·K, where the discriminant has a negative sign with a positive determinant, indicating that they are stable points with complex and conjugated eigenvalues. These points are characterized by a local behavior in the space of the phases determined by self-oscillatory trajectories.

Having studied the possibilities that determine the geometry of the valve, we found two possible situations where the slope of the characteristic curve can be negative. The first one, when the valve was mounted in a reverse position, and the second one, at high temperatures, because of saturation problems, the coolant flow decreased when the temperature increased. The first of these situations could only occur in a specific device configuration design. The second can be given in practical applications where there is a problem of flow saturation, for example, in the coolant distributor.

## 4. Conclusions

This study presents a thermal analysis of a temperature-driven microfluidic cell using a nonlinear self-adaptive micro valve that provides the mechanisms for the system to efficiently maintain a critical temperature. With some simplifying assumptions, the problem was described by a set of two ordinary differential equations (cell model equations) subjected to one boundary condition. The model equations defined are linear, and the nonlinearity of the system lies exclusively on the boundary condition describing the buckling behavior of the self-adaptive valve. The dimensionless transcendent equation that defines the valve behavior was described by an exponential curve with an error (RMSE) lower than 1.6%.

The parameters of the microfluidic cell model were calibrated so that the stationary solution would be identical to the obtained using the PDE model for different values of heat flux and coolant. System behavior was simulated with a 0.5% error in the description of stationary points and a 1.0% error in dynamic temporal behavior with 4 changes in system conditions, which reflect a wide range of situations founded in laboratory.

As the characteristics of the cell itself can introduce instabilities in the behavior of the thermal system, which can be aggravated when operating in cell matrix configuration, the effect of a microfluidic cell, with the self-adaptive thermostatic valve, in the stationary solutions of the system for different working conditions was investigated. The characteristics stationary diagrams obtained were based on one non-linear branch, showing how the temperature stabilizes with an increase of the heat flux and the strong nonlinearity between the thermal resistance of the device and the heat flux applied. This effect is the key for its use in cooling applications; ultimately, the temperature of the valve regulates the effective thermal resistance of the device, ranging from 1.60 × 10^−5^ to 2.0 × 10^−4^ K m^2^/W.

The stability analysis was performed by studying the linear perturbation around the stationary solution. The eigenvalues of the linear matrix, representing the resultant transformation, determined the type of stability in every case. The model was solved for various heat flows, critical flow rates, and temperatures. The main conclusions are listed below:The stationary points with complex eigenvalues and negative real parts were stable with damped oscillatory evolution. These types of points represent the first step to destabilization along the temporal evolution of the system.The intervals for the different design parameters for which the device had an asymptotically stable stationary solution were identified. They corresponded to the set of conditions necessary to avoid possible stability problems.A general study over the parameter space established that instabilities (saddle-node bifurcations) only appear if the slope of the self-adaptive valve characteristic curve is negative and lower than 0.01 m/s·K. Within this range, a specific valve design would be required.

## Figures and Tables

**Figure 1 micromachines-12-00505-f001:**
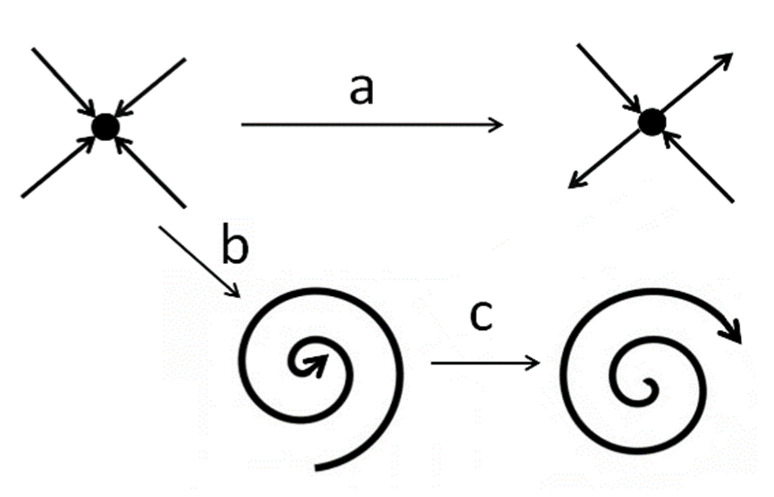
Schematic representation of the types of system behavior around the stationary points for several control parameter values. Different routes are noun by a, b, and c.

**Figure 2 micromachines-12-00505-f002:**
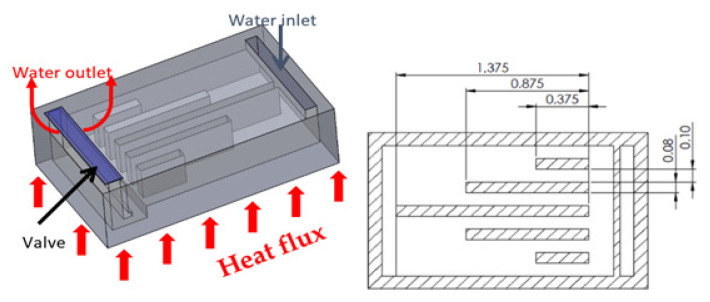
Microfluidic cell with self-adaptive valve and micro channels inside with micro channels dimensions (in mm).

**Figure 3 micromachines-12-00505-f003:**
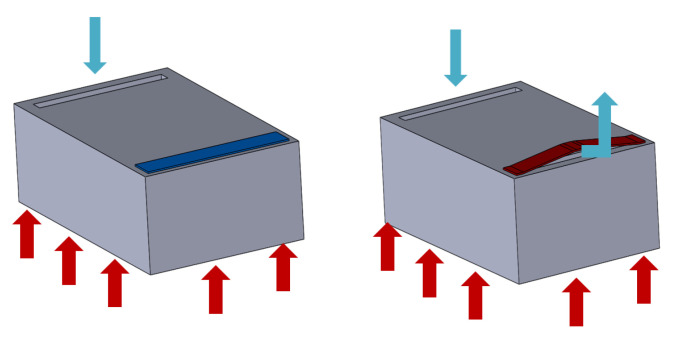
Microfluidic cell with the micro valve and its buckling movement.

**Figure 4 micromachines-12-00505-f004:**
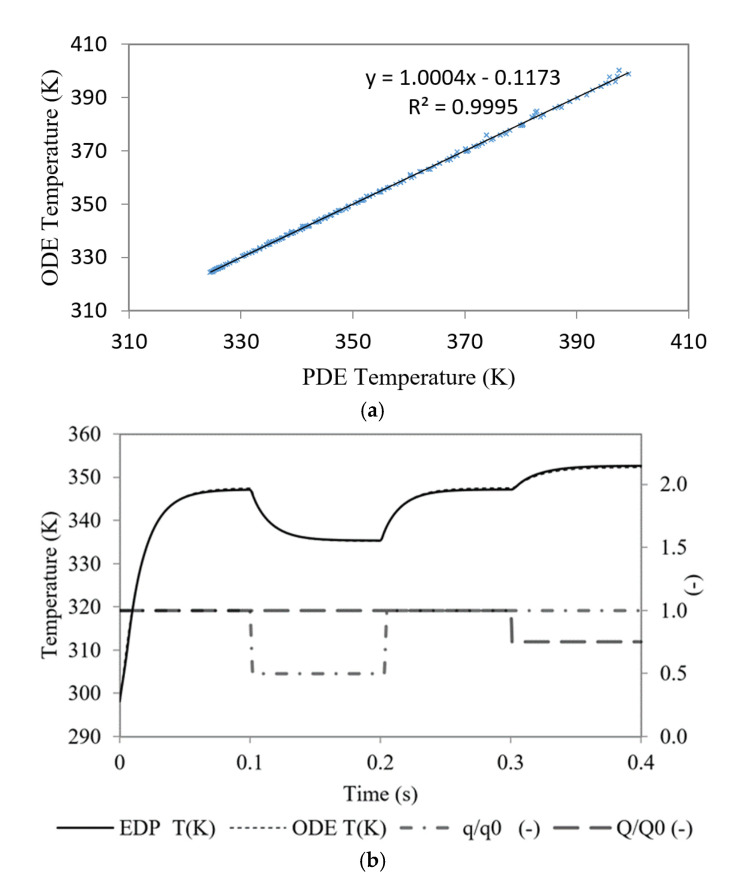
Comparison of (**a**) stationary solution calculated with proposed microfluidic cell model (ODE) and partial differential equation model (PDE) and (**b**) dynamical solution with 4 different transitions: *t* = 0 s initial heat up (q = 300 W/cm^2^ and flow rate Q_0_ = 3.0 mL/min), *t* = 0.1 s change to q = 150 W/cm^2^, *t* = 0.2 s again q = 300 W/cm^2^, and *t* = 0.3 reduction of the coolant flow rate Q = 3/4 Q_0_.

**Figure 5 micromachines-12-00505-f005:**
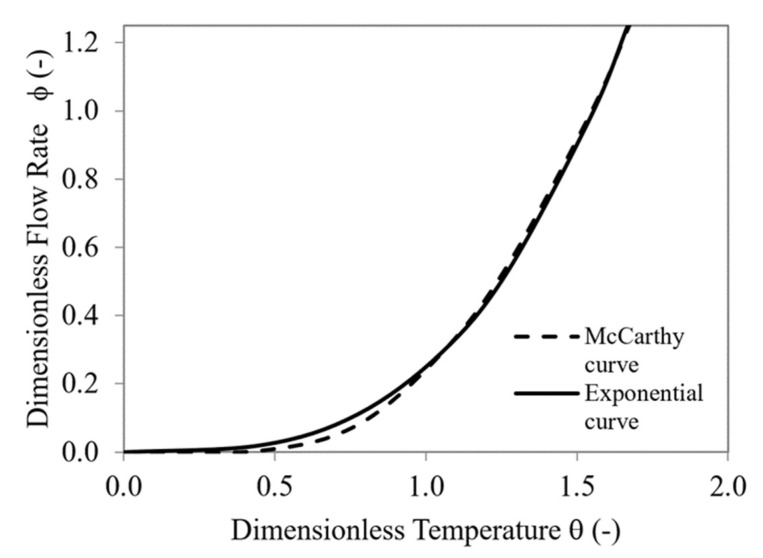
Characteristic valve curve. Comparison between McCarthy curve and exponential curve (Equation (7)).

**Figure 6 micromachines-12-00505-f006:**
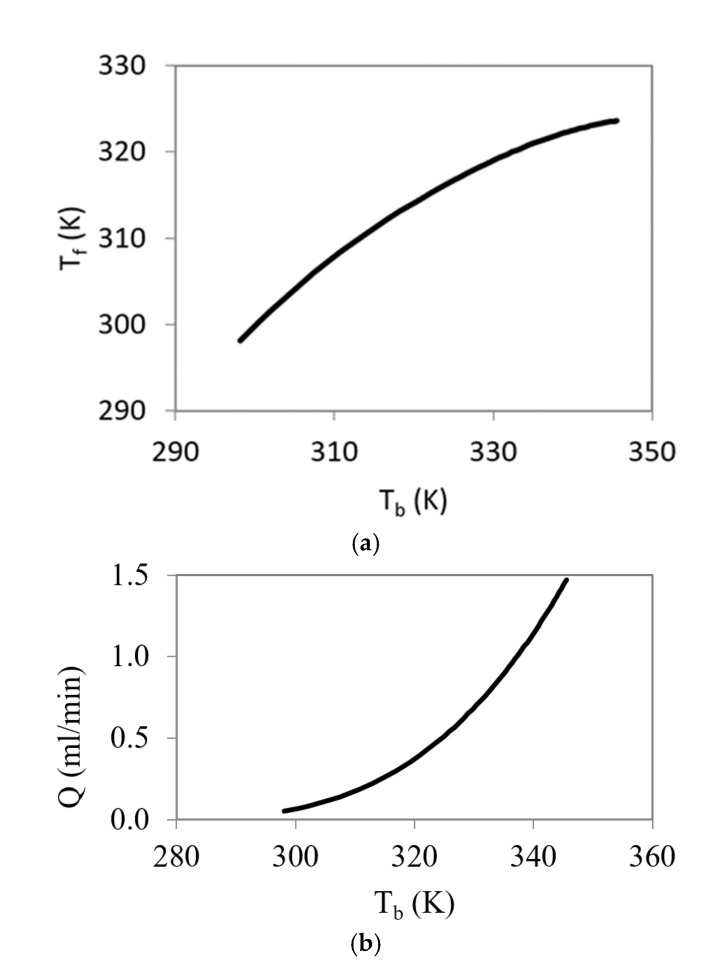
Relation between T_b_ and T_f_ in stationary solution for different heat flux values (**a**) and flow rate versus cell temperature (**b**).

**Figure 7 micromachines-12-00505-f007:**
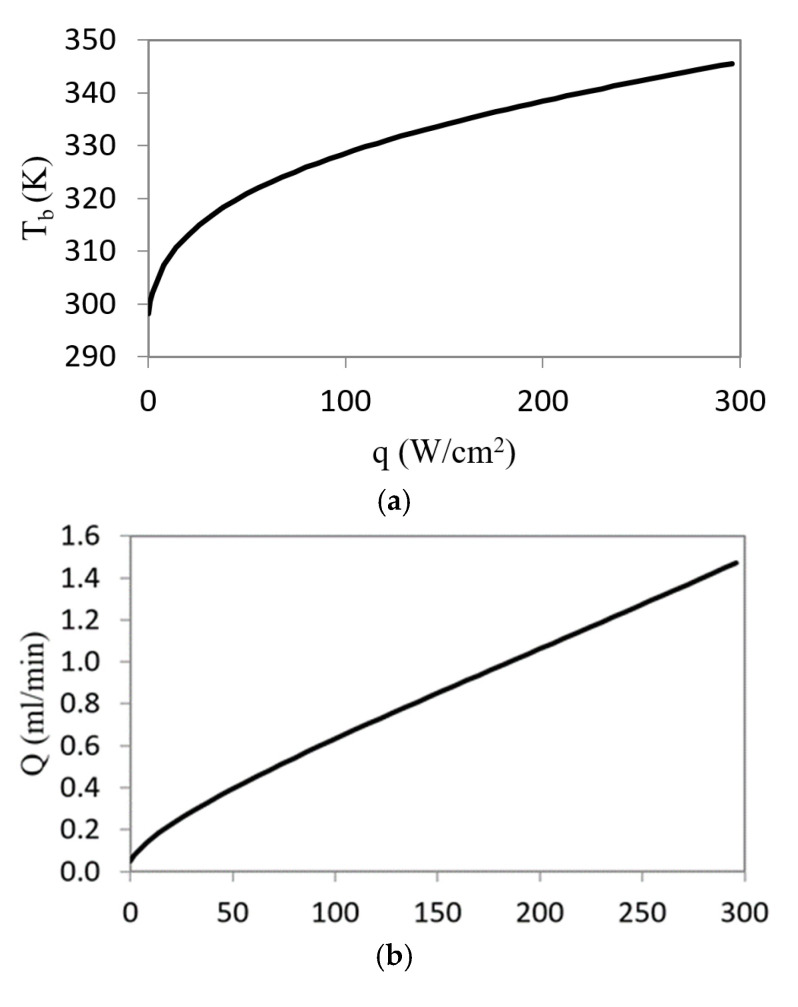
Microfluidic cell temperature (**a**) flow rate; (**b**) and thermal resistance; (**c**) versus heat flux.

**Figure 8 micromachines-12-00505-f008:**
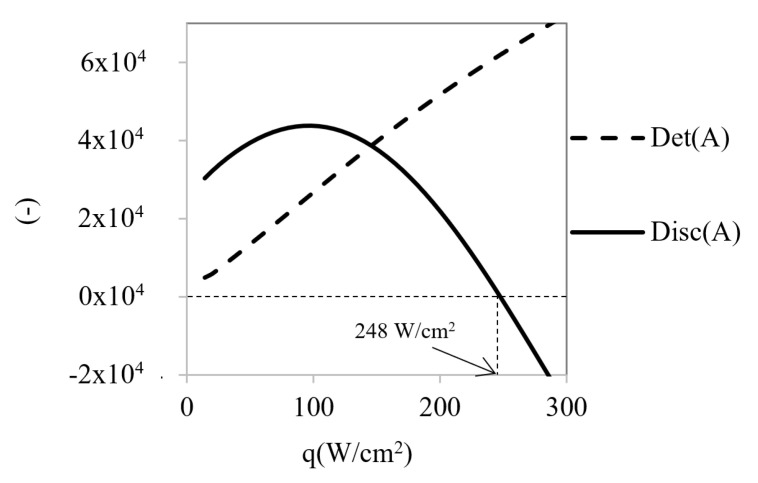
Determinant and discriminant values for different heat flow rates.

**Figure 9 micromachines-12-00505-f009:**
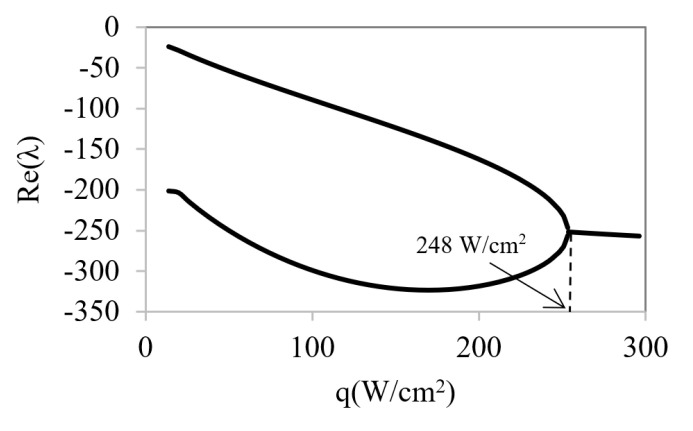
Real part of eigenvalues λ_1_ and λ_2_ as a function of heat flux.

**Figure 10 micromachines-12-00505-f010:**
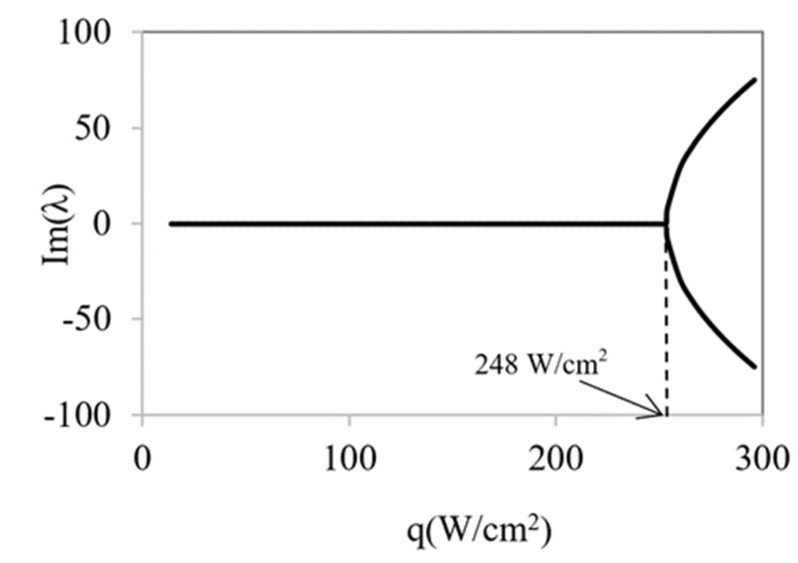
Imaginary part of eigenvalues λ_1_ and λ_2_ as a function of heat flux.

**Figure 11 micromachines-12-00505-f011:**
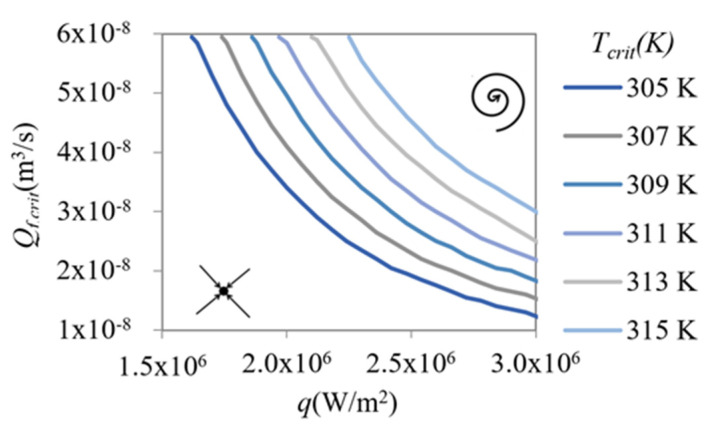
Boundary line of the self-oscillating behavior zone based on the design parameters (Q*_f,crit_, T_crit_*,) and heat flux (*q*).

**Figure 12 micromachines-12-00505-f012:**
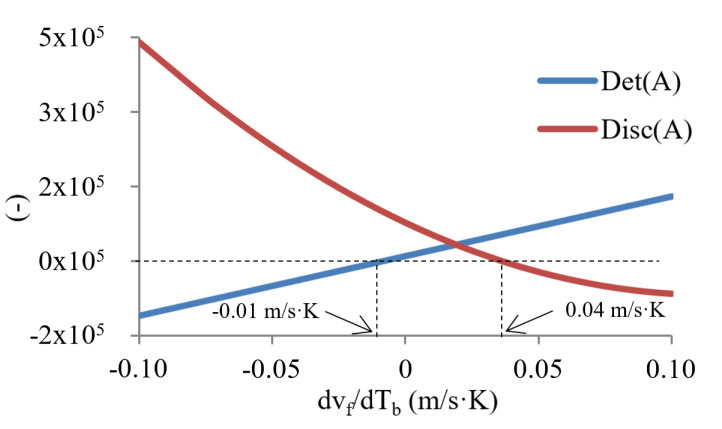
Determinant and discriminant for different valve equation slopes.

**Figure 13 micromachines-12-00505-f013:**
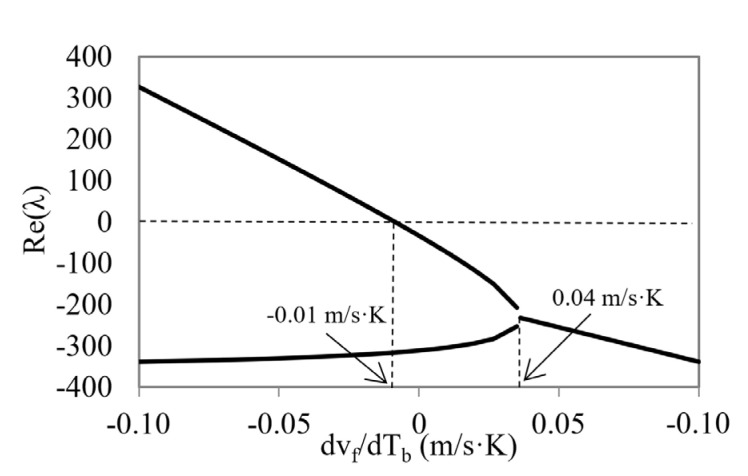
Real part of λ_1_ and λ_2_ eigenvalues as a function of velocity derivative.

**Table 1 micromachines-12-00505-t001:** Values used to calculate stationary solution.

$C_{b}$ (J/K·m^2^)	654.17	$C_{f}$ (J/K·m^2^)	846.13
$U_{L}\;$(W/K·m^2^)	20.1	*S* ($m^{2})$	5.0$\times 10^{-8}$
$T_{crit}$ (K)	320.0	$T_{a}$ (K)	298.15
$T_{ref}\;$(K)	273.15	$T_{in}$ (K)	298.15
$Q_{f}{}_{crit}$ (mL/min)	2.5		

**Table 2 micromachines-12-00505-t002:** Parameters of calibration.

$c_{1}\;$ (s^3^/J·K)	$1.94\times 10^{-8}$	$m_{2}\;$ (J/K·m^3^)	$3.02\times 10^{5}$
$c_{2}\;$ (s^2^/K·m^2^)	$-1.24\times 10^{-1}$	$r_{1}$ (J/s·K·m^2^)	$4.21\times 10^{4}$
$c_{3}$ (J·s/K·m^4^)	$-2.09\times 10^{5}$	$n_{1}$ (J/K·m^3^)	$2.38\times 10^{5}$
$m_{1}\;$ (s/K·m)	$2.92\times 10^{-2}$	$n_{2}$ (J/K·s·m^2^)	$-9.19\times 10^{2}$

**Table 3 micromachines-12-00505-t003:** Parameters of valve Equation (7).

$K_{1}$ (-)	τ (-)	*ex* (-)
$8.85\times 10^{3}\;$	$2.8\times 10^{-5}$	3.15

## Data Availability

The data presented in this study are available upon reasonable request from the authors.

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
