# Peer review of "Thermal Analysis of a MEMS-Based Self-Adaptive Microfluidic Cooling Device"

_micromachines, 2021, doi:10.3390/mi12050505_

Round 1
Reviewer 1 Report
This paper addresses a numerical study on a temperature-driven microfluidic cell through a nonlinear self-adaptive micro valve to keep the system under critical conditions at an efficient way. The topic is relevant. The manuscript is overall well written, the results are well discussed and support the main conclusions. My main concerns are with the details on the model validation and on the numerical method, that should be described with more detail. Here is a list of specific comments that I would like to see addressed in a revised version of the manuscript: Fig. 2: the overall dimensions of the device are given, but the dimensions of the channels are not clear and must be. The parameter G defined in eq 4 is like a heat transfer parameter, why does it have a value between 1 and 2? It is not clear how the model is validated. It is not clear which mesh is used, the size and number of elements, the numerical domain, the boundary conditions…. these information must be detailed.Author Response
Please see the attachment.

Reviewer 2 Report
- This paper reports on a thermal analysis of microfluidic cell to maintain a critical temperature. The analysis is well organized and the theoretical background is clearly summarized. To improve the paper quality, the following comments are suggested.
- L 112, what is the difference between equation (1) and (2)?
- L 113, the term Ub (convective heat coefficient) is generally represented as ‘h’ and ‘k’ for conduction. If possible, follow the common notation.
- L129, Need reference for the constants.
- There are series of related equations (up to 23) in detail. Try to shorten the derivation of equations.
- L239, in Fig. 6, subscripts need to be described as subscript form.
- L294, color lines in the figure needs to described as dotted or dashed for the clear identification.
- L334, Figure 23, what are red underlines?
- L385, Nomanclature needs to be moved to right after the introduction section.
Round 2
Reviewer 1 Report
Authors have revised the paper taking into account the major comments from the reviewer. The paper is suitable now